# MARGINAL BENEFIT INDUCED UNSUPERVISED ENVIRONMENT DESIGN

## ABSTRACT

Training generally capable Reinforcement Learning (RL) agents in complex environments is a challenging task that involves designing appropriate distributions of environments. Recent research has highlighted the potential of the Unsupervised Environment Design (UED) framework, which generates environments at the frontier of the agent's capabilities through adaptive curriculum learning using a *regret*-based objective. Regret-based approaches have been used for either controlled (e.g., PAIRED) or random (e.g., PLR) generation of environments. While regret-based approaches have shown great promise in generating feasible environments, they can produce difficult environments that are challenging for the agent to learn from. This is because regret represents the best-case learning potential of an environment, without indicating how much the agent can actually learn from it. To that end, we propose an alternative objective that employs marginal benefit, focusing on the improvement in the agent policy associated with the environment. This new objective generates environments at a suitable pace for the agent's learning and thus achieves rapid convergence. More importantly, we provide a closed-loop controlled environment generation approach (similar to PAIRED) that employs Marginal Benefit and a new notion of environment diversity. Finally, we provide detailed experimental results and ablation analysis to showcase the effectiveness of our new methods.

## 1 INTRODUCTION

The advancements in Reinforcement Learning (RL) have led to significant successes in various applications, such as game playing (Mnih et al., 2015; Silver et al., 2016), robot control (Levine et al., 2016; Akkaya et al., 2019), and many others. However, training RL agents with general capabilities remains a major challenge due to the millions of experiences required to train an RL agent in each environment, which is both time-consuming and expensive.

One promising approach to address this problem is to train an agent in a "shallow" manner on a sequence of tasks or environments (Dennis et al., 2020; Jiang et al., 2021b; Parker-Holder et al., 2022; Li et al., 2023). In this method, instead of extensively training on each environment with millions of experiences, the student agent is exposed to a limited number of experiences in each individual environment. This adaptive curriculum of environments, tailored to the student's policy, has been demonstrated to produce more robust agents in fewer training steps (Portelas et al., 2020a; Jiang et al., 2021b). This methodology is referred to as Unsupervised Environment Design (UED).

Protagonist Antagonist Induced Regret Environment Design (PAIRED (Dennis et al., 2020)) introduced a self-supervised RL paradigm in which the RL teacher employs regret, obtained from the student's performance to generate new environments. This approach leverages regret as a measure of learning potential to create environments that are at the edge of the capabilities of the student. However, computing the regret value, which is approximated by the difference between the expected payoffs of the student (protagonist) and the expert (antagonist), is computationally expensive owing to costly interactions between the three agents and the environment. Furthermore, PAIRED suffers from catastrophic forgetting of past environments due to learning on new environments. To address these concerns, subsequent works such as PLR (Jiang et al., 2021b) and ACCEL (Parker-Holder et al., 2022) introduced multiple changes: (1) They eliminated RL-based generation and instead employed a random generation of environments; (2) They used an approximate version of

regret, namely Generalized Advantage Estimate (GAE). (3) They employed the replay of past environments to prevent catastrophic forgetting. (4) They edited the generated environments to ensure thorough training on a challenging sequence of environments (as an RL generator would do).

In this paper, we first provide an alternative measure for the fundamental measure used by the teacher to generate environments (either through RL or random methods), i.e., regret or its variant, GAE. While regret represents the learning potential of an environment, it fails to indicate whether all that potential can be achieved by a student and if so in how many steps. Therefore, we propose an alternative measure for environment generation called marginal benefit. Marginal benefit quantifies the actual improvement obtained by the student due to "training" on an environment. In order to be sample efficient in large scale problems, we believe RL based controlled generation of environments is more suitable if the approach is scalable. Therefore, we provide an approach for RL based controlled generation of environments using marginal benefit as the measure.

**Contributions:** Towards operationalizing the ideas of utilizing marginal benefit-based measure and RL based controlled generation, we make the following key contributions:

- We define the marginal benefit measure for an environment and use it in a teacher (RL-based or random) to generate environments. More importantly, we propose an RL-based teacher that is more scalable compared to the original RL generator UED algorithm, PAIRED.

- To facilitate generality in training with marginal benefit, we define a new notion of "diversity" in environments and propose a mechanism to intentionally train on "diverse" environments with high marginal benefit.

- We demonstrate the effectiveness of our methods through extensive experiments on a wide range of benchmark problems. We are able to achieve SOTA results through RL-based generation.

## 2 BACKGROUND

In this section, we provide a brief background on UED and discuss relevant methods for UED.

### 2.1 UNSUPERVISED ENVIRONMENT DESIGN, UED

In UED, we train a student to perform well across a set of in-distribution and out-of-distribution environments. To accomplish this, UED utilizes a teacher agent that provides a sequence of environment parameter values to train the student to generalize well to unseen levels. The UED problem is formally described as an Underspecified Partially Observable Markov Decision Process (UPOMDP):

$$\langle \mathbb{S}, \mathbb{A}, \mathbf{\Omega}, \boldsymbol{\theta}, T, R, O, \gamma \rangle$$

$\mathbb{S}$, $\mathbb{A}$ and $\mathbf{\Omega}$ are the set of states, actions, and observations, respectively. $R : \mathbb{S} \to \mathbb{R}$ is the reward function, and $\gamma$ is the discount factor. The crucial element is $\boldsymbol{\theta}$, which denotes the set of all possible environment configurations. A particular parameter configuration, $\boldsymbol{\theta} \in \boldsymbol{\theta}$ (can be a vector or sequence of values) defines a level and can impact the reward model, transition dynamics, and the observation function, i.e. $R : \mathbb{S} \times \boldsymbol{\theta} \to \mathbb{R}$, $T : \mathbb{S} \times \mathbb{A} \times \boldsymbol{\theta} \to \mathbb{S}$ and $O : \mathbb{S} \times \boldsymbol{\theta} \to \mathbf{\Omega}$. The UPOMDP is underspecified because training with all values of $\boldsymbol{\theta}$ ($\in \boldsymbol{\theta}$) is infeasible, as $\boldsymbol{\theta}$ can be infinitely large. The goal of the student policy $\pi$ in a UPOMDP is to maximize its discounted expected rewards for any given $\boldsymbol{\theta} \in \boldsymbol{\theta}$. In the model-free setting, where transition and observation functions are not known *a priori*, this objective is formulated as:

$$\max_{\pi} V^{\boldsymbol{\theta}}(\pi) = \max_{\pi} \mathbb{E}_{\pi}\Big[ \sum_{t=0}^{H} r_t^{\boldsymbol{\theta}} \cdot \gamma^t \Big]$$

where $r_t^{\boldsymbol{\theta}}$ is the reward obtained by the student policy $\pi$ in a level with environment parameter $\boldsymbol{\theta}$ at time step $t$. Consequently, the student needs to be trained on a series of $\boldsymbol{\theta}$ values that maximize its generalization capability across all possible levels from $\boldsymbol{\theta}$. To achieve this objective, the teacher agent is employed. The goal of the teacher policy is to generate a distribution over the next set of environment parameter values to train the student, i.e.,$\mathbf{\Lambda} : \mathbf{\Pi} \to \Delta(\boldsymbol{\theta})$ to achieve good generalization performance, where $\mathbf{\Pi}$ is the set of possible policies of the teacher.

## 2.2 Existing methods

Dennis et al. (2020) first formalized the UED and introduced the Protagonist Antagonist Induced Regret Environment Design (PAIRED) algorithm, which is a three-agent game: the protagonist $\pi^P$ (student), the antagonist $\pi^A$ (expert) and the environment generator $\mathcal{G}$ (teacher). The environment generator $\mathcal{G}$ learns to control the distribution of environmental parameters $\boldsymbol{\theta}$ by maximizing regret, which is approximated by the difference between the cumulative reward obtained by the protagonist and the antagonist under the same environment with parameters $\boldsymbol{\theta}$:

$$\text{REGRET}^{\boldsymbol{\theta}}(\pi^P, \pi^A) = V^{\boldsymbol{\theta}}(\pi^A) - V^{\boldsymbol{\theta}}(\pi^P) \tag{1}$$

Both the protagonist and antagonist are trained to maximize their own cumulative reward in the current environment $\boldsymbol{\theta}$. Note that the environment generator (teacher) is discouraged from generating levels that can not be solved because they will have a maximum regret of 0. This teacher-student-expert framework co-evolves the policies, creating an adaptive curriculum learning approach where the teacher constantly creates an emergent class of levels that get progressively more difficult along the borderline of the student's ability, allowing agents to learn a good policy that enables zero-shot transfer. More specifically, if $\boldsymbol{\Pi}$ is the strategy set of the protagonist and antagonist, and $\boldsymbol{\theta}$ is the strategy set of the teacher, then if the learning process reaches a Nash equilibrium, the resulting student policy $\pi$ provably converges to a minimax regret policy, defined as

$$\pi = \arg\min_{\pi^P \in \boldsymbol{\Pi}}\{\arg\max_{\boldsymbol{\theta}, \pi^A \in \boldsymbol{\theta}, \boldsymbol{\Pi}}\{\text{REGRET}^{\boldsymbol{\theta}}(\pi^P, \pi^A)\}\} \tag{2}$$

However, this framework struggles because of teacher efficiency, as it uses a RL generator and requires expensive interactions with the environment to collect millions of samples to train Protagonist and Antagonist agents separately.

As an alternative regret-based UED approach, Jiang et al. (2021b) proposed Prioritized Level Replay (PLR), where a student policy is challenged by two co-evolving teachers, Generator and Curator. In PLR, Generator randomly creates new environments, while Curator prioritizes the replay probability for each environment based on the estimated learning potential. By adapting the sampling of the previously encountered levels to train, PLR is an active learning strategy that improves sample efficiency and generalization. In addition, PLR uses Generalized Advantage Estimation (GAE) (Schulman et al., 2015) to approximate the regret compared to the more expensive regret definition used in PAIRED. Specifically, the regret $F_{gae}(\boldsymbol{\theta})$ of the environment $\boldsymbol{\theta}$ is defined as:

$$F_{gae}(\boldsymbol{\theta}) = \frac{1}{T}\sum_{t=0}^{T}\max\{\sum_{k=t}^{T}(\gamma\boldsymbol{\Lambda})^{k-t}\delta_k, 0\} \tag{3}$$

where $\boldsymbol{\Lambda}$ and $\gamma$ are the exponential weight discount and MDP discount factor respectively. $\delta$ is the TD-error at timestep $t$. The agent trained by PLR shows good generalization ability in terms of empirical results, and PLR has further been deployed at large scale domains (Bauer et al., 2023). However, PLR is still limited as it is unable to exploit any previously discovered level structure and can only curate randomly sampled levels. Moreover, the random search will be heavily affected by the high-dimensional design space, making it highly unlikely to sample levels at the frontier of the agent's current capabilities.

## 3 Algorithm: *MBeDED*

In this section, we propose the Marginal Benefit and Diversity based Environment Design (*MBe-DED*) approach to address UED problems. Our algorithm, *MBeDED* , relies on the teacher-student framework, which consists of an environment generator (teacher) and two student agents which help compute the marginal benefit of a generated environment. Algorithm 1 provides the pseudocode of the overall algorithm, and Figure 1 provides the overall framework. The *MBeDED* algorithm incorporates two key components that differentiate it from existing research in solving UED problems

- Marginal benefit-based environment generator that can either be controlled (RL based) or randomized.
- Diversity guided environment selection

**Algorithm 1** *MBeDED*

1: **Input:** Level buffer $\mathbf{\Lambda}$, replay probability $p$.
2: **Initialize**: policy Alice $\pi^A$, Bob $\pi^B$, level genera-
   tor $\mathcal{G}$ and overall replay distribution $P_{replay}$;
3: **while** Not converged **do**
4:     Sample a replay decision, $\epsilon \sim U[0,1]$
5:     **if** $\epsilon \geq p$ **then**
6:         Generate $\boldsymbol{\theta}$ from $\mathcal{G}$, and create POMDP $\mathcal{M}_{\boldsymbol{\theta}}$
7:         Collect Alice's and Bob's trajectories $\tau^A$
           and $\tau^B$ in $\mathcal{M}_{\boldsymbol{\theta}}$, and compute $V^{\boldsymbol{\theta}}(\pi^A) =$
           $\sum_{t=0}^{T} \gamma^t r_t^{\boldsymbol{\theta}}$ and $V^{\boldsymbol{\theta}}(\pi^B) = \sum_{t=0}^{T} \gamma^t r_t^{\boldsymbol{\theta}}$
8:         Compute $\mu$ using Eq. 4
9:         Update $\pi^B$ by letting $\pi^B = \pi^A$
10:        Train $\pi^A$ to maximize $V^{\boldsymbol{\theta}}(\pi^A)$
11:        Update $\mathcal{G}$ with RL using reward $\mu(\pi^A, \pi^B)$
12:        Determine $\boldsymbol{S_{\theta}}$
13:        If $F_{div}(,)$ of $\boldsymbol{\theta}$ is higher than the lowest one
           in the $\mathbf{\Lambda}$, replace that one by $\boldsymbol{\theta}$ to $\mathbf{\Lambda}$ and up-
           date $P_{replay}$ according to Eq. 13
14:    **else**
15:        Sample $\boldsymbol{\theta}$ from $\mathbf{\Lambda}$ according to $P_{replay}$, and
           create POMDP $\mathcal{M}_{\boldsymbol{\theta}}$
16:        Collect Alice's and Bob's trajectories $\tau^A$
           and $\tau^B$ in $\mathcal{M}_{\boldsymbol{\theta}}$, and compute $V^{\boldsymbol{\theta}}(\pi^A) =$
           $\sum_{t=0}^{T} \gamma^t r_t$ and $V^{\boldsymbol{\theta}}(\pi^B) = \sum_{t=0}^{T} \gamma^t r_t$
17:        Update $\pi^B$ by letting $\pi^B = \pi^A$
18:        Train $\pi^A$ to maximize $V^{\boldsymbol{\theta}}(\pi^A)$
19:        Update $\boldsymbol{S_{\theta}}$
20:        Update $P_{replay}$ according to Eq. 13
21:    **end if**
22: **end while**

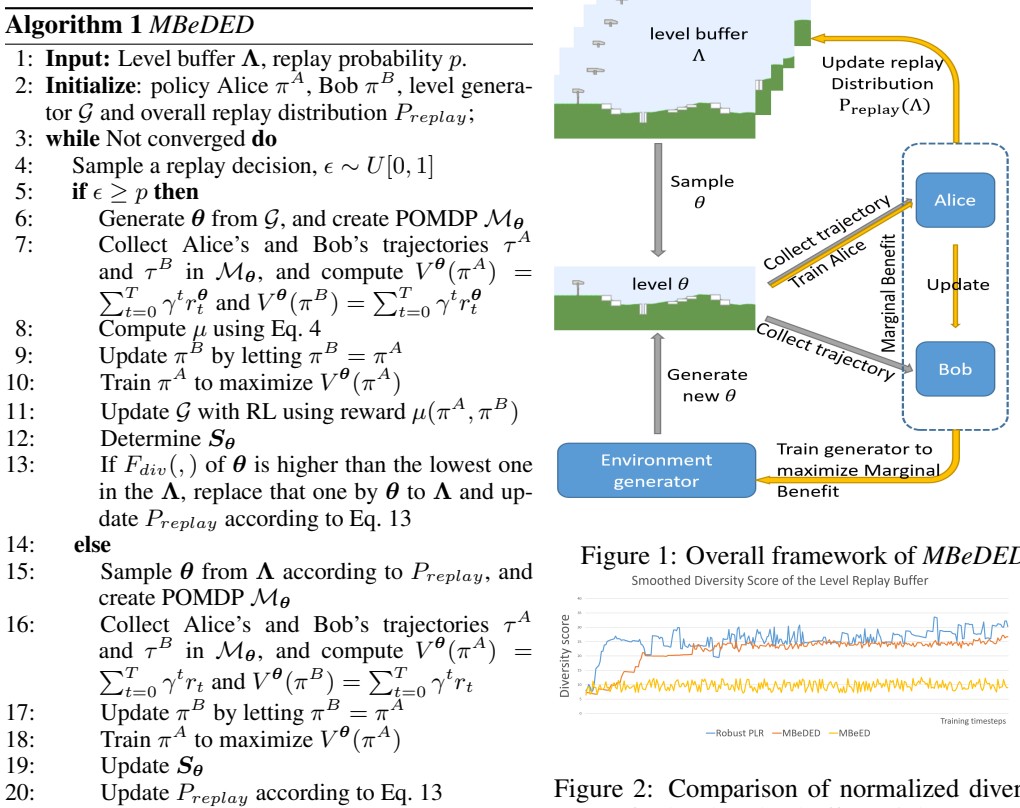

Figure 1: Overall framework of *MBeDED*

Figure 2: Comparison of normalized diversity scores for level replay buffer. Higher scores indicate higher diversity.

*MBeDED* incentivizes the teacher to automatically generate challenging levels that push the agent to its limits by utilizing the concept of marginal benefit. To compute the marginal benefit, two versions of the student, Alice and Bob, are maintained. Alice represents the student with the latest policy, while Bob holds a slightly outdated policy. We define Marginal Benefit as the difference in values between policies of Alice and Bob. For marginal benefit to be positive, the teacher needs to generate environments that improve the policy. This serves as the incentive for the teacher to keep generating environments that improve the student policy.

### 3.1 MARGINAL BENEFIT BASED RL ENVIRONMENT GENERATOR

We first describe the marginal benefit-based RL generator (teacher). The teacher utilizes the marginal benefit computed from the student policies for the generated environment, to guide the learning process. To compute the marginal benefit ($\mu$) of a generated environment ($\boldsymbol{\theta}$), we calculate the difference in the expected value obtained by Alice and Bob on that environment:

$$\mu^{\boldsymbol{\theta}}(\pi^A, \pi^B) = V^{\boldsymbol{\theta}}(\pi^A) - V^{\boldsymbol{\theta}}(\pi^B) \tag{4}$$

Both Alice and Bob will collect their trajectories, denoted as $\tau^A$ and $\tau^B$ respectively, in the current level $\boldsymbol{\theta}$ (Line 6 and 15 in Algorithm 1). The marginal benefit of training on the previous environment is then computed as the difference between the cumulative rewards they received (Line 7).

There are alternative methods available to compute the marginal benefit, such as using the difference in the scoring function based on the average magnitude of the Generalized Advantage Estimate (GAE; (Schulman et al., 2015)) over each of the $T$ time steps. However, in this work, we opted to compute the marginal benefit as the difference between Bob and Alice's expected cumulative returns because of its simplicity and promising results in creating challenging yet solvable environments. By focusing on improvement in student policy for a generated environment, we adopt a student-

centric perspective, which allows for generating environments at a suitable pace and provides a more accurate representation of the student's actual improvement during training.

In order to generate a challenging environment at a suitable pace, the environment generator (teacher) is trained to maximize the marginal benefit when creating a new environment (Line 10). A crucial aspect of this approach is the direct policy copying from Alice to Bob (Line 8 and Line 16), which avoids the need for costly interactions with the environment to train an optimal antagonist's policy at the current level, as required in PAIRED (Dennis et al., 2020)). Then Alice is trained to maximize the corresponding cumulative reward (Line 9 and Line 17). The self-regulating feedback loop between Alice and Bob enables the environment generator to establish an adaptive curriculum, where new levels that could significantly alter Alice and Bob's behavior are constantly generated.

## 3.2 DIVERSITY GUIDED ENVIRONMENT SELECTION

The objective of this section is to incorporate diversity into the UED framework to enhance effective explorations and improve the generalizability of the student agent. Existing algorithms ((Jiang et al., 2021a;b; Parker-Holder et al., 2022)) that utilize random generation also consider a setup whereby a level buffer $\mathbf{\Lambda}$ is introduced to store the top $K$ visited levels with the highest learning potential. The learning potential is estimated by the GAE value of the student agent over the last episode.

The key intuition behind introducing diversity is that while the marginal benefit-induced RL generation is a good criterion for generating an environment, it may not contribute to generalization if the level replay buffer $\mathbf{\Lambda}$ contains many similar environments. In Figure 2, we show the normalized diversity score calculated by our proposed diversity measurement. Random-based generation methods such as Robust PLR encounter fewer similarities due to their inherent randomness. However, RL-based generation is more likely to consistently produce similar or redundant environments over time. Specifically, the *MBeED* approach, which focuses solely on marginal benefit, records the lowest diversity compared to *MBeDED* and Robust PLR. Determining the level buffer by the learning potential alone may result in preserving similar or repeated levels, resulting in low-quality exploration, and the agent will not learn much from training on these similar environments.

Therefore, we propose to selectively add diverse environments to the level buffer, so as to ensure that the student agent gets exposure to a wider variety of environments (as shown in Figure 2, *MBe-DED* can achieve the same diversity level compared to random generation approaches). To that end, we introduce a *Diversity* measure that can be beneficial for UED approaches. One possible measure of diversity between two levels/environments $\boldsymbol{\theta}_1$ and $\boldsymbol{\theta}_2$ can be the distance between their parameters. However, this method fails to capture the stochasticity in the mapping from parameters to the environment. For example, in the continuous-control *BipedalWalker* environment, the environment design space is an 8-dimensional indirect encoding representing the intensity of four kinds of obstacles for the student agent. The actual positioning of obstacles is random, meaning that the same parameter values can yield different environments. Therefore, this simple distance measure is inadequate for measuring diversity.

Instead, we focus on measuring the diversity (negative similarity) of representative observed states corresponding to environments when the student policy is executed on the environment.

### 3.2.1 REPRESENTATIVE OBSERVED STATE VECTOR

We begin by defining the representative observed state vector, $\boldsymbol{S_\theta}$, which corresponds to an environment given a fixed student policy.

**Definition 1** *(**Representative observed state vector**) For an environment $\boldsymbol{\theta}$, given multiple trajectories collected by the current student policy, we have the set of all visited states, $\mathbb{S} = \{\boldsymbol{s}_1, \boldsymbol{s}_2, ...\boldsymbol{s}_n\}$. We define the representative observed state vector, $\boldsymbol{S_\theta}$ as*

$$\boldsymbol{S_\theta} = \{\tilde{\boldsymbol{s}}_1, \tilde{\boldsymbol{s}}_2, ....\tilde{\boldsymbol{s}}_m\}, \text{ where } m \ll n$$

*and $\boldsymbol{S_\theta} \subset \mathbb{S}$ represents the set of representative states visited in the trajectories.*

$\boldsymbol{S_\theta}$ consists of two types of states:

- **important** states: we rank the observed states according to their TD-error, $\delta^A = r_t + \gamma V(\boldsymbol{s}_{t+1}) - V(\boldsymbol{s}_t)$, which is also used in the GAE computation. States with high TD errors are considered

important as they have a significant impact on training. We select the top $m_1$ states with the highest TD error and include them in $\boldsymbol{S_\theta}$.

- **representative** states: In addition to important states, we aim to ensure that $\boldsymbol{S_\theta}$ effectively represents the environment $\boldsymbol{\theta}$. This means that for every observed state in $\mathbb{S}$, there should be a similar state in $\boldsymbol{S_\theta}$. We add the remaining $m_2 = m - m_1$ states to ensure that $\boldsymbol{S_\theta}$ provides a comprehensive representation of the level $\boldsymbol{\theta}$.

Formally, the representative score of $\boldsymbol{S_\theta}$ is defined as:

$$F_{rep}(\boldsymbol{S_\theta}) = \sum_{\boldsymbol{s}_i \in \mathbb{S}} \max_{\boldsymbol{s}_j \in \boldsymbol{S_\theta}} \{k(\boldsymbol{s}_i, \boldsymbol{s}_j)\} \tag{5}$$

where $k(.,.)$ represents the similarity kernel between states. One possible choice for the similarity kernel is the cosine similarity, defined as:

$$k(\boldsymbol{s}_1, \boldsymbol{s}_2) = \frac{\boldsymbol{s}_1 \boldsymbol{s}_2^\top}{\|\boldsymbol{s}_1\|\|\boldsymbol{s}_2\|}, \tag{6}$$

where $\|\boldsymbol{s}\|$ represents the norm of observed state $\boldsymbol{s}$. It is important to note that if $k(\boldsymbol{s}_1, \boldsymbol{s}_2) = 1$, it implies that $\boldsymbol{s}_1$ and $\boldsymbol{s}_2$ are identical.

While finding the important states is simple, determining the $\boldsymbol{S_\theta}$ that maximizes $F_{rep}(.)$ is NP-hard, and it is computationally expensive when $\mathbb{S}$ is typically very large. Motivated by Fang et al. (2019), we propose a heuristic way. We first randomly sample a set $\boldsymbol{S}' \subset \mathbb{S}$ of size $n'$, where $m < n' < n$. Then, we use a greedy algorithm to pick the top $m_2$ observed states from $\boldsymbol{S}'$. we start by initializing $\boldsymbol{S_\theta}$ as a set including those top $m_1$ important states, and at each step, we will add the observed state $\boldsymbol{s}$ that maximizes the marginal gain to $\boldsymbol{S_\theta}$, where the marginal gain $F_{rep}(\{\boldsymbol{s}\}|\boldsymbol{S_\theta})$ is defined as the difference in $F_{rep}(\cdot)$ when adding the observed state $\{\boldsymbol{s}\}$ to $\boldsymbol{S_\theta}$:

$$F_{rep}(\{\boldsymbol{s}\}|\boldsymbol{S_\theta}) = F_{rep}(\{\boldsymbol{s}\} \cup \boldsymbol{S_\theta}) - F_{rep}(\boldsymbol{S_\theta}) \tag{7}$$

Because $F_{rep}(\boldsymbol{S_\theta})$ is a submodular function, the greedy algorithm can provide a solution, $\boldsymbol{S_\theta^*}$ with an approximation factor of $1 - \frac{1}{e}$ (Nemhauser et al., 1978).

### 3.2.2 DIVERSE LEVEL REPLAY BUFFER

We now describe how the representative observed state vector is utilized to maintain a diverse level replay buffer (utilized in Algorithm 1). A diverse level buffer is more informative and will contribute to the effective exploration and improve the generalizability of the agent. The formal definition to measure the state-aware diversity among the level buffer $\boldsymbol{\Lambda}$ for a given student policy is as follows.

**Definition 2** (*Diversity score among* $\boldsymbol{\Lambda}$) *Consider the level replay buffer* $\boldsymbol{\Lambda} = \{\boldsymbol{\theta}_1, \ldots, \boldsymbol{\theta}_K\}$, *each level* $\boldsymbol{\theta}_i \in \boldsymbol{\Lambda}$ *has its corresponding representative observed state vector,* $\boldsymbol{S}_{\boldsymbol{\theta}_i}$. *We can compute the state-aware diversity score* $F_{div}(\boldsymbol{\Lambda}, \boldsymbol{\Lambda})$ *of level replay buffer* $\boldsymbol{\Lambda}$ *as follows:*

$$F_{div}(\boldsymbol{\Lambda}, \boldsymbol{\Lambda}) = \sum_{\boldsymbol{\theta}_i \in \boldsymbol{\Lambda}} F_{div}(\boldsymbol{\theta}_i, \boldsymbol{\Lambda} \setminus \{\boldsymbol{\theta}_i\}) \tag{8}$$

$F_{div}(\boldsymbol{\theta}_i, \boldsymbol{\Lambda} \setminus \{\boldsymbol{\theta}_i\})$ *measures the diversity score between the level* $\boldsymbol{\theta}_i$ *and a set of levels* $\boldsymbol{\Lambda} \setminus \{\boldsymbol{\theta}_i\}$ *and*

$$F_{div}(\boldsymbol{\theta}_i, \boldsymbol{\Lambda} \setminus \{\boldsymbol{\theta}_i\}) = - \sum_{\boldsymbol{s}_i \in S_{\boldsymbol{\theta}_i}} \max_{\boldsymbol{s}_j \in \{S_{\boldsymbol{\theta}_j}\}_{i \neq j}} \{k(\boldsymbol{s}_i, \boldsymbol{s}_j)\} \tag{9}$$

Unlike computing the representative score in Eq. 5, we introduce a negative sign before the kernel function to assess the diversity between two states. A high similarity between two representative observed state vectors indicates a low diversity score between those two levels, and vice versa. It is important to note that this diversity measure between levels is not symmetric, as $F_{div}(\boldsymbol{\theta}_i, \boldsymbol{\Lambda} \setminus \{\boldsymbol{\theta}_i\}) \neq F_{div}(\boldsymbol{\Lambda} \setminus \{\boldsymbol{\theta}_i\}, \boldsymbol{\theta}_i)$. We can make it symmetric by letting $F_{div}^{sym} = \frac{F_{div}(\boldsymbol{\theta}_i, \boldsymbol{\Lambda} \setminus \{\boldsymbol{\theta}_i\}) + F_{div}(\boldsymbol{\Lambda} \setminus \{\boldsymbol{\theta}_i\}, \boldsymbol{\theta}_i)}{2}$. However, since the symmetric property has no impact on our algorithm, we use the asymmetric form in this work. When generating a new environment $\boldsymbol{\theta}_{new}$, if that new level is added to the level replay buffer $\boldsymbol{\Lambda}$, we want to increase the diversity among $\boldsymbol{\Lambda}$, which means the diversity score among $\boldsymbol{\Lambda}$,

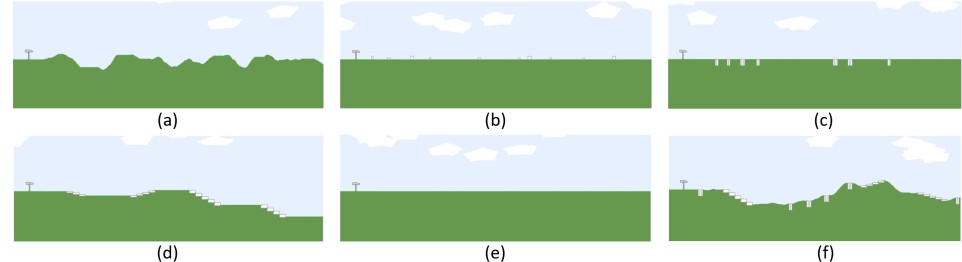

Figure 3: An illustration of levels generated with four kinds of obstacles: (a) Roughness of range (2,8) (b) Stump height of range (1,3) (c) Pit gap of range(1,3) (d) Stair steps of range (2,6) (e) Vanilla *BipedalWalker* (f)Hard *BipedalWalker* with a mix of (a) to (d) parameters.

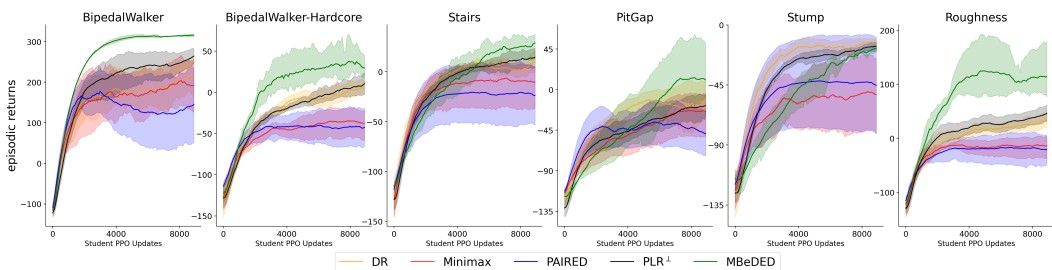

Figure 4: Transfer performance on test environments during training (mean and standard error).

i.e., $F_{div}(\Lambda, \Lambda)$, should increase. We now provide a heuristic way to determine whether a newly generated level $\theta_{new}$ should be added to the level replay buffer $\Lambda$:

[1] Consider the level buffer $\Lambda = \{\theta_1, \ldots, \theta_K\}$ and the newly generated level $\theta_{new}$, each of which has its corresponding observed state representative vector $S_\theta$. For any level $\theta \in \{\theta_{new}\} \cup \Lambda$, we can compute its state-aware diversity score $F_{div}(,)$ with other levels $\theta' \in \{\theta_{new}\} \cup \Lambda \setminus \{\theta\}$ as follows:

$$F_{div}(\theta, \{\theta_{new}\} \cup \Lambda \setminus \{\theta\}) = - \sum_{o_i \in S_\theta} \max_{o_j \in S'_\theta} \{k(o_i, o_j)\}, \tag{10}$$

[2] If $F_{div}(\theta_{new}, \Lambda)$ is higher than the $\min_{\theta_i \in \Lambda} \{F_{div}(\theta_i, \{\theta_{new}\} \cup \Lambda \setminus \{\theta_i\})\}$, add $\theta_{new}$ to $\Lambda$ and remove $\theta_i$ that has the lowest $F_{div}(\theta_i, \{\theta_{new}\} \cup \Lambda \setminus \{\theta_i\})$ value, as a higher diversity score of $\theta_{new}$ indicates a lower likelihood of finding a similar observed state within the levels in $\Lambda$.

### 3.3 LEVEL REPLAY

At the beginning of each iteration, *MBeDED* either generates new levels (with probability $p$, line 4 and 5) or sample a mini-batch of levels in the level buffer to train the student (line 13 and 14). Details regarding the replay process are explained in the appendix.

## 4 EXPERIMENTAL RESULTS

In this section, we present our experimental results in the domains of *BipedalWalker*, *Minigrid*, and *CarRacing* to demonstrate the superior performance of our approach when a trained agent is transferred to new environments. We compare our approach against existing UED methods: Domain Randomization (DR (Tobin et al., 2017)), Minimax (Wang et al., 2019), PAIRED (Dennis et al., 2020), PLR (Jiang et al., 2021a). We do not compare our approach against ACCEL (Parker-Holder et al., 2022) as it re We show the average and variance of the performance for our method, baselines with five random seeds. There are other random generation UED employing techniques complementary to the above baselines. Table 1 in the appendix provides a summary of the key characteristics of all approaches.

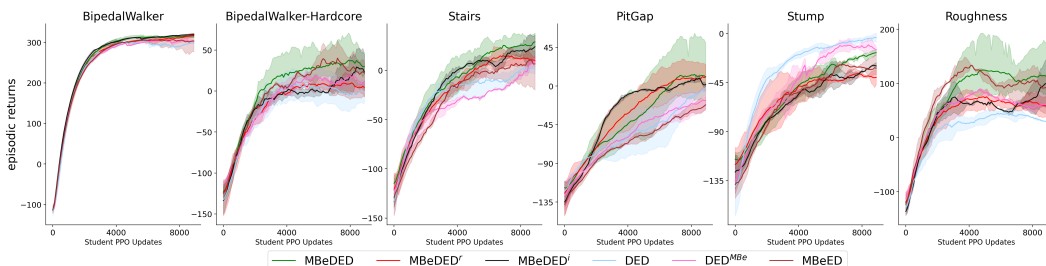

Figure 5: The results of the ablation experiment on the transfer performance of the agent during training (mean and standard error).

**Performance on BipedalWalker:** We first evaluate our approach on *BipedalWalker* (Wang et al., 2019) environment. This environment entails continuous control with dense rewards. Similar to Wang et al. (2019), we use a modified version of *BipedalWalker-Hardcore* from OpenAI Gym (Brockman et al., 2016). In *BipedalWalker*, there are 8 parameters that indirectly represent the intensity of four kinds of terrain-based obstacles for a two-legged robot: the minimum/maximum roughness of the ground, the minimum/maximum height of stump obstacles, the minimum/maximum width of pit gap obstacles, and the minimum/maximum size of ascending and descending flights of stairs. We provide an illustration of these four kinds of obstacles in Figure 3.

The *BipedalWalker* environment provides the student agent with a 24-dimensional proprioceptive state with respect to its lidar sensors, angles, and contacts. The action space is continuous and consists of four values that control the torques of the agent's four motors. In this environment, the teacher learns to control eight parameters that correspond to the range of four kinds of obstacles and then combines a random seed to generate a specific level. All agents are trained using Proximal Policy Optimization (PPO, (Schulman et al., 2017)). For a fair comparison, during training, we presented a vanilla *BipedalWalker*, a challenging *BipedalWalker-Hardcore* environment, and four specific levels in the context of isolated challenges in {Roughness, Stump height, Pit gap, Stair step} to evaluate our algorithm and baselines.

Figure 4 shows the transfer performance throughout training. As shown in the figure, *MBeDED* consistently outperforms all the baselines across the majority of test environments, achieving faster convergence (and is more scalable compared to the original RL generator algorithm, PAIRED). These results provide strong evidence in support of the key principles driving *MBeDED*'s design: a student-centric generator that progressively produces environments to facilitate the improvement of the student policy, and the maintenance of diverse environments to enhance effective exploration and improve generalizability.

We also provide ablation analysis to evaluate the impact of different design choices, including (a) RL-based generator vs. Random generator; (b) important states vs. representative states; (c) marginal benefit vs. regret-based incentive for generator. The results are provided in Figure 5. Specifically, *MBeDED$^r$* considers only representative states when selecting $S_\theta$ for level $\theta$, while *MBeDED$^i$* considers only the important observed states. *DED* and *DED$^{MBe}$* are both versions of our algorithm without an RL-based environment generator. Similar to PLR, they use a randomly generated environment teacher, and DED approximates the learning potential using GAE, *DED$^{MBe}$* approximates the learning potential using the marginal benefit of level $\theta$. As we see, in the parameterized *BipedalWalker* environment, the RL-based teacher can effectively exploit previously discovered level structure or adapt the difficulty of the environment to the student's learning progress.

**Performance on Minigrid:** Here we investigate the maze navigation environment introduced by Dennis et al. (2020), which is based on *Minigrid* (Chevalier-Boisvert et al., 2018). We train the environment generator to learn how to build maze environments by choosing the location of the obstacles, the goals, and the starting location of the agent. Specifically, at the beginning of each iteration, the generator will place the student agent and the goal, and then every time step afterward, the generator outputs a location where the next obstacle will be placed. There will be up to 50 blocks that can be placed. Several examples of generated mazes during training are illustrated in Figure 6.

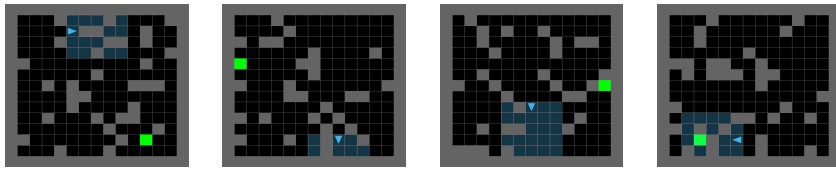

Figure 6: Example levels generated by *MBeDED* (placing up to 50 blocks) during training.

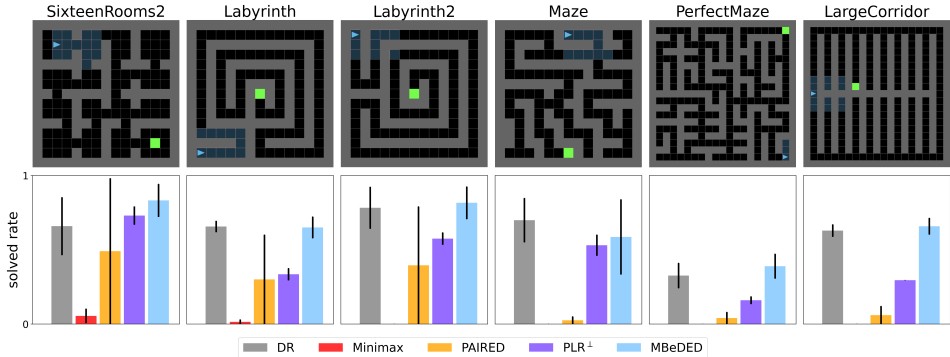

Figure 7: Zero-shot transfer performances in challenging environments after 100 million training steps. We show the median and interquartile range of solved rates over 5 runs.

The maze is partially observable, where the student agent's view is shown as a blue-shaded area in Figure 6. The student agent (blue triangle) must explore the maze to find a goal (green square). In order to deal with the partially observable setting, our agents use PPO with a Recurrent Neural Network structure. We compare our agents' transfer ability trained by different approaches on human-designed levels. The test environments and the performance are reported in Figure 7. While DR acts as a strong baseline in this domain, *MBeDED* can achieve a similar highest mean return.

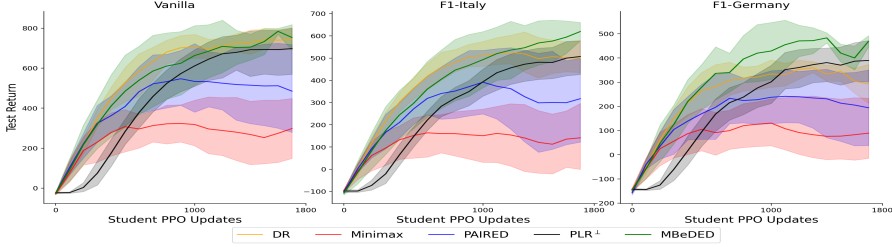

Figure 8: Zero-shot transfer performance on the OOD F1 tracks: Vanilla, Italy and Germany.

The results of the ablation study on the Maze environment are provided in the appendix.

**Performance on Car Racing:** Figure 8 provides the results for the car racing domain, and *MBeDED* is able to outperform other baselines. All details of the experiments can be found in the appendix.

## 5 CONCLUSION

In this paper, we introduce *MBeDED* , a novel method for unsupervised environment design. Our approach utilizes a curriculum to automatically create a distribution of training environments by employing a novel marginal benefit-based measure. Furthermore, in order to enhance effective exploration and improve generalizability, *MBeDED* selectively revisits previously generated environments by prioritizing those with higher estimated learning potential and diversity. Finally, we conducted experiments in various benchmark environments and demonstrated that our RL-based generation algorithm achieves superior zero-shot transfer performance in most settings.

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

## A    APPENDIX

## B    RELATED WORK

This work aims to train agents that are capable of generalizing across a wide range of environments (Whiteson, 2009). Several methods for enhancing generalization in RL utilize techniques from supervised learning, such as data augmentation (Raileanu et al., 2020; Kostrikov et al., 2020; Wang et al., 2020), and feature distillation (Igl et al., 2020). In contrast to supervised learning, there is a growing trend of incorporating curriculum learning mechanisms in various learning scenarios (Fang et al., 2019; Weinshall & Amir, 2020; Wu et al., 2020). In RL, curricula improve the learning performance of the agent by adapting the training environment to the agent's current capabilities. One prior approach is domain randomization (Jakobi, 1997; Tobin et al., 2017), where agents are trained on a wide range of randomly generated environments. In contrast, Akkaya et al. (2019) propose automatic domain randomization, where they use a curriculum that gradually increases the difficulty of agent training. In the multi-task domain (Sukhbaatar et al., 2017; Zhang et al., 2020; Du et al., 2022; Klink et al., 2022), automatic curricula are specifically designed for the goals that agents need to solve. These Curricula are often generated as the proposed goals are right at the frontier of the learning process of an agent.

In particular, we focus on the emerging field of unsupervised environment design (Dennis et al., 2020), which is inherently related to the Automatic Curriculum Learning (Florensa et al., 2017; Portelas et al., 2020b). It seeks to learn a curriculum that adaptively generates challenging environments to train robust agents. Dennis et al. (2020) proposed PAIRED algorithm, where they introduce an environmental adversary that learns a curriculum to control environmental parameters to maximize approximate regret. POET (Wang et al., 2019; 2020) co-evolves the generation of environmental challenges and the optimization of agents to solve them. (Jiang et al., 2021b;a) further introduce PLR, a general framework that allows agents to revisit previously generated environments with high learning potential for training. We draw inspiration from these works, leveraging automatically generated environments with a curriculum design and maintaining a level buffer with high learning potential and diversity.

For brevity, we provide the works most related to our approach and summarize them in Table 1.

Table 1: Key Characteristics of Baselines and *MBeDED*. MCC is an abbreviation for Minimal Criteria Coevolution (Wang et al., 2019).

| Algorithm | Generation Strategy | Generator Objective | Buffer Objective | Setting |
|---|---|---|---|---|
| DR (Tobin et al., 2017) | Random | None | None | Single Agent |
| Minimax (Wang et al., 2019) | Evolution | Minimax | MCC | Population-Based |
| PAIRED (Dennis et al., 2020) | RL | Minimax Regret | None | Single Agent |
| PLR (Jiang et al., 2021a) | Random | None | Learning potential | Single Agent |
| *MBeDED* | RL | Maximize Marginal Benefit | Diversity & Learning Potential | Single Agent |

## C    ADDITIONAL DETAILS IN ALGORITHM

### C.1    DIVERSITY GUIDED ENVIRONMENT SELECTION

Note that the objective of the diversity score for the level buffer is different from that of finding the representative observed state for levels. In the case of the representative state, a high diversity score is desirable as it ensures that every observation obtained from the level can find a similar state in the set $S_\theta$. However, for the level buffer, we aim to maintain diversity, which implies minimal similarity. Therefore, a higher diversity score of new environment $\theta_{new}$ indicates a lower likelihood of finding a similar observed state in the levels within the level buffer.

When considering the use of either important states or representative states to represent the environment $\theta$, determining the important states is straightforward. However, determining the representative states is more complex. Intuitively, we use the cosine similarity kernel $k$ to measure how well the

selected observed states in the set $S_{\boldsymbol{\theta}}$ can represent the whole observed states $\mathbb{S}$ collected from the current level. A high representative score $F_{rep}(S_{\boldsymbol{\theta}})$ indicates that each observed state collected from the current level can find a sufficiently similar state in $S_{\boldsymbol{\theta}}$, such that $S_{\boldsymbol{\theta}}$ is a good representation of the current level. The way to find the $S_{\boldsymbol{\theta}}$ only based on the representative states is as follows:

1. We first randomly sample a set $S' \subset \mathbb{S}$ of size $m'$, where $n < m' \ll m$;
2. Then we use a greedy algorithm to pick the top $n$ observed states from the set $O'$. we start by taking $S_{\boldsymbol{\theta}}$ as an empty set and at each step, we will add the observed state $s$ that maximizes the marginal gain, where the marginal gain $F_{rep}(\{s\}|S_{\boldsymbol{\theta}})$ is defined as the difference of adding observed state $\{s\}$ into $S_{\boldsymbol{\theta}}$:

$$F_{rep}(\{s\}|S_{\boldsymbol{\theta}}) = F_{rep}(\{s\} \cup S_{\boldsymbol{\theta}}) - F_{rep}(S_{\boldsymbol{\theta}}) \qquad (11)$$

## C.2 Level replay

At the beginning of each iteration, *MBeDED* either generates new levels (with probability $p$, line 4 and 5) or sample a mini-batch of levels in the level buffer to train the agent (line 13 and 14).

- (**Generating new level**): When there is a newly generated level, we measure the diversity score of the new level and the levels in the level buffer according to Equation 10. To achieve a diverse level buffer, if the diversity score of a new level is lower than one of the levels in the buffer, we add the new level $\boldsymbol{\theta}_{new}$ to the buffer $\boldsymbol{\Lambda}$ to replace the level with the highest diversity score $F_{div}(,)$ (Line 11 and 12).
- (**Sampling level from buffer**): In order to decide which level to train on, we assign each level $\boldsymbol{\theta}_i$ a probability that is based on the combination of its diversity score and learning potential. Following Jiang et al. (2021b), we use the GAE function shown in Equation 3 as the proxy for its learning potential. Given the learning potential of $F_{gae}(\boldsymbol{\theta}_i)$, we rank them accordingly and use a prioritization function $h$ to decide how differences in learning potential are translated into differences in prioritization. As a result, we obtain a learning potential prioritized distribution $P_{gae}(\boldsymbol{\Lambda})$ over the level buffer, and the probability for $\boldsymbol{\theta}_i$ is

$$P_{gae}(\boldsymbol{\theta}_i|\boldsymbol{\Lambda}, F_{gae}) = \frac{h\left(\mathrm{rank}(F_{gae}(\boldsymbol{\theta}_i))\right)^{1/\beta}}{\sum_j h\left(\mathrm{rank}(F_{gae}(\boldsymbol{\theta}_j))\right)^{1/\beta}} \qquad (12)$$

where $\beta$ is the temperature parameter that tunes how much $h(\mathrm{rank}(F_{gae}(\boldsymbol{\theta}_i)))$ determines related probability, and $\mathrm{rank}(F_{gae}(\boldsymbol{\theta}_i))$ is the rank of level $\boldsymbol{\theta}_i$ sorted in the descending order among the level buffer. Same to Jiang et al. (2021b), we employ $h\left(\mathrm{rank}(F_{gae}(\boldsymbol{\theta}_j))\right) = \frac{1}{\mathrm{rank}(F_{gae}(\boldsymbol{\theta}_i))}$. Note that different surrogates for learning potential can be used, such as the marginal benefit $\mu(\pi^A, \pi^B)$ shown in Equation 4. The comparison of using different types of learning potential is provided in the following section in the Appendix.

Similarly, we can compute the diversity score computed through Equation 10 and its corresponding diversity score prioritized distribution $P_{div}(\boldsymbol{\Lambda})$ over the level buffer. We also rank $F_{div}(,)$ in descending order, as we prefer to train on environments that are more informative and contain diverse observed states. We update the overall replay distribution $P_{replay}(\boldsymbol{\Lambda})$ over $\boldsymbol{\Lambda}$ by combining $P_{gae}(\boldsymbol{\Lambda})$ based on the learning potential and $P_{div}(\boldsymbol{\Lambda})$ based on the diversity score as (Line 19):

$$P_{replay}(\boldsymbol{\Lambda}) = (1 - \rho) \cdot P_{gae}(\boldsymbol{\Lambda}) + \rho \cdot P_{div}(\boldsymbol{\Lambda}) \qquad (13)$$

The hyperparameter $\rho \in [0, 1]$ is used to balance the trade-off between learning potential and diversity. Therefore, a level that exhibits a higher learning potential or a greater diversity in its observed states is more likely to be selected for replaying. We present the overall framework for *MBeDED* in Figure 1.

## D Additional experiments

### D.1 Other possible approaches

We compare *MBeDED* with other different UEDs during training by periodically evaluating them on the test environments. We include a comparison with DIPLR (Li et al., 2023), a recent approach that incorporates Diversity for environment generation. Their approach involves evaluating

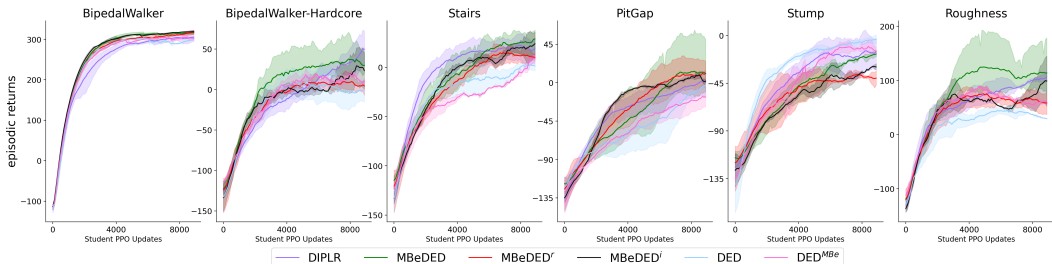

Figure 9: Performance on test environments during training (mean and standard error). Here The DIPLR method by Li et al. (2023) randomly generate new levels. *MBeDED*$^r$ considers only representative states when selecting $\boldsymbol{S_\theta}$ for level $\boldsymbol{\theta}$, while *MBeDED*$^i$ considers only the important observed states. DED and DED$^{MBe}$ are both versions of our algorithm without the self-play component. Similar to PLR or DIPLR, they use a randomly generated environment teacher, and DED approximates the learning potential using GAE, DED$^{MBe}$ approximates it using the marginal benefit of level $\boldsymbol{\theta}$.

the similarity on different levels based on the distance between occupancy distributions of the current student policy. They decide whether to include a new level in the buffer using a combination of this distance measure and regret value. To increase diversity, they add levels with the highest distance value to the level buffer. In contrast, our diversity measurement is more scalable than DIPLR approach. DIPLR utilizes state-action occupancy measures and computes the diversity through the Wasserstein distance between these occupancy measures. They will update occupancy measures for all environments in the level buffer once the student agent policy is updated with the currently selected environment. The whole process of this step is very time-consuming as they need to re-collect trajectories in all environments in the level replay buffer (as shown in the DIPLR paper) and the computation of the Wasserstein distance is computationally intensive. However, our diversity measurement is more efficient by only requiring updates to the representative observed states vector of the currently selected environment. Additionally, our diversity measurement demonstrates comparable results in some experiments, such as the DED approach (random-based generation with our diversity measurement) displaying better performance against DIPLR in some test environments across different domains.

## D.2 BIPEDAL WALKER

we also conduct ablation studies in the Bipedal walker environment to determine which factor influences performance most. The result is shown in Figure 9.

As we can see, our proposed algorithm consistently outperforms DIPLR. Particularly, the combination of the important state and representative state as the representative observed state vector exhibits the best performance across most cases. In addition, we evaluate the transfer performance of alternative approaches for approximating learning potentials in the BipedalWalker environment. It is important to note that in this evaluation, we used a random generation as the environment generator (teacher). This comparison is shown in section D.5 and Figure 13.

## D.3 MINIGRID

We also conducted an ablation study in the Minigrid domain, comparing our approach against the concurrent work by Li et al. (2023). The results of this comparison are presented in the figure 10.

Furthermore, we present the results for the scenario where an RL-based generator is not used, instead, we use the random generated teacher. These results are shown in Figure 12.

## D.4 PERFORMANCE ON CARRACING

Finally, we investigate the learning dynamics of *MBeDED* and baselines on *CarRacing* (Brockman et al., 2016), a popular continuous-control environment with dense rewards. Similar to the partially

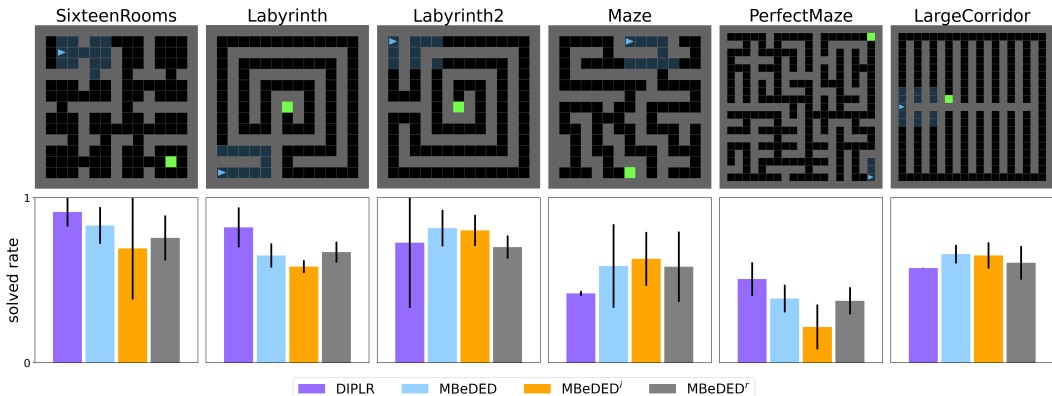

Figure 10: Zero-shot transfer performances in challenging environments after 100 million training steps. We show the median and interquartile range of solved rates over 5 runs.

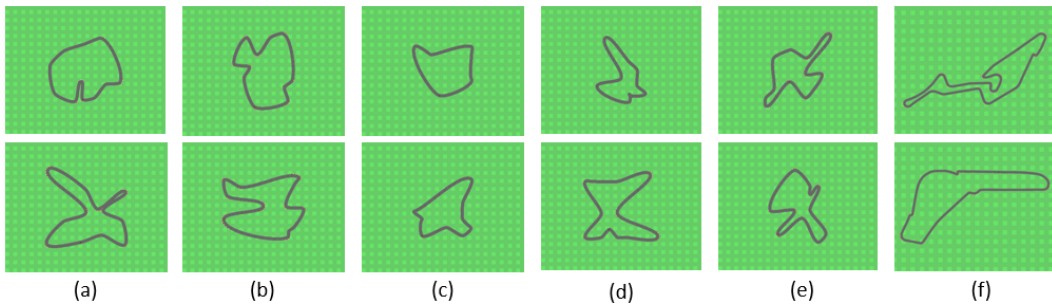

Figure 11: A randomly-selected examples of *CarRacing* tracks produced by different algorithms. (a)DR (b) Minimax (c) PAIRED (d) PLR (e)*MBeDED* (f) Two examples in the *CarRacing* F1 benchmark that are used for evaluating zero-shot generalization.

observable navigation task, The student agent in *CarRacing* receives a partial, pixel observation and has a 3-dimensional action space. The goal of the agent is to drive a full lap around a generated track. To generate a feasible level (closed-loop track), following (Jiang et al., 2021a), the generator learns to choose a sequence of up to 12 control points, which will unique generate a Bézier curve (Mortenson, 1999) within predefined curvature constraints. In Figure 11, we show some examples of *CarRacing* tracks produced by different algorithms.

We present per-track zero-shot transfer returns of policies trained by each method on some of the human-designed Formula One (F1) tracks throughout training in Figure 8. Note that these tracks are significantly out-of-distribution (OOD) as they can not be generated within 12 control points. Remarkably, *MBeDED* can either mitigate the degeneracy of PAIRED or achieve significant outperformance than other baselines in mean performance, providing further evidence of the benefits of the induced curriculum and diverse level buffer. We also present the comparison between our approach and DIPLR in the provided Figure 15. The results clearly demonstrate that MBeDED is capable of achieving comparable performance to DIPLR in the given task.

## D.5 LEARNING POTENTIAL

In this section, we conduct experiments with additional methods for approximating the learning potential of an environment, aiming to effectively sample replayed levels with high learning potentials. As per the arguments in section 3.1, we define the marginal benefit of environment $\theta$ as the difference in performance in the next environment after training the student agent in the environment $\theta$ compared to the expected performance of the student in the same next environment $\theta'$ before train-

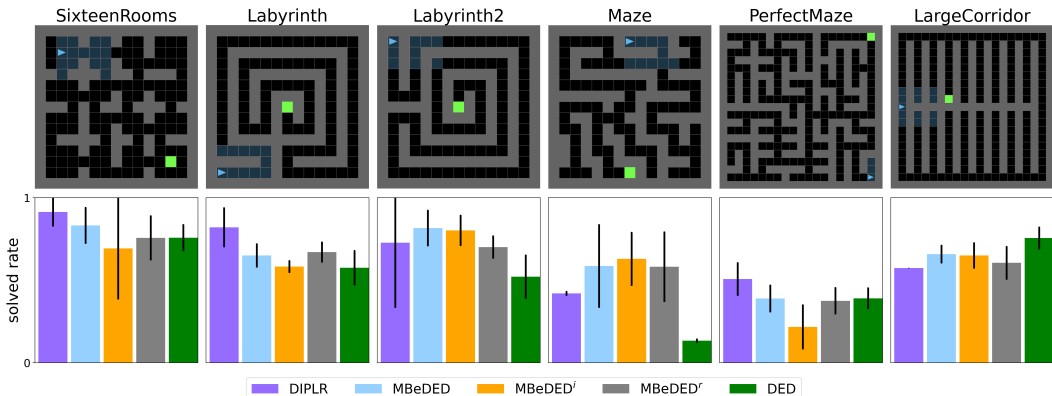

Figure 12: Zero-shot transfer performances in challenging environments after 100 million training steps. We show the median and interquartile range of solved rates over 5 runs.

ing. This can be interpreted as the learning potential of the environment with respect to the student's learning. We rewrite the two different definitions to approximate the learning potential here for convenience:

- PLR uses the *positive value loss*(referred to as GAE) to approximate the regret, and further is used as the learning potential in the level replay buffer. Specifically, regret is approximated by:

$$F_{gae}(\boldsymbol{\theta}) = \frac{1}{T} \sum_{t=0}^{T} \max\{\sum_{k=t}^{T} (\gamma\boldsymbol{\Lambda})^{k-t}\delta_k, 0\} \tag{14}$$

- In this work, we can also approximate the learning potential by the marginal benefit which is defined as the difference between the cumulative reward received by Alice $\pi^A$ and Bob $\pi^B$:

$$\mu^{\boldsymbol{\theta}}(\pi^A, \pi^B) = V^{\boldsymbol{\theta}'}(\pi^A) - V^{\boldsymbol{\theta}'}(\pi^B) \tag{15}$$

As we can easily have this value after collecting Alice $\pi^A$ and Bob $\pi^B$'s trajectories in the next environment $\boldsymbol{\theta}'$, we can compute the marginal benefit $\mu^{\boldsymbol{\theta}}(\pi^A, \pi^B)$ according to Equation 15 and assign this value to the previous training environment $\boldsymbol{\theta}$ as the learning potential. Therefore, we developed an alternative approach using the marginal benefit. One potential disadvantage of this method is that the calculation of the learning potential for a given environment $\boldsymbol{\theta}$ is delayed. Instead of directly calculating the potential in the current environment like using the GAE function, we obtain the potential of the previous environment $\boldsymbol{\theta}$ from the current environment $\boldsymbol{\theta}'$, which may result in instability. Because the value obtained can be influenced by the current environment. As a result, different next environments may lead to different learning potentials for previous environments, which can cause instability.

We provide the experiment results for the BipedalWalker domain. Figure 13 compares the transfer performance of both methods with the RL-based environment generator (teacher) component removed, using different methods for calculating regret (using GAE function as shown in Equation 3 and marginal benefit as shown in Equation 4).To isolate the impact of different methods for approximating learning potentials, we generate new environments using a random teacher. Both methods demonstrate reasonable transfer performance and maintain similar abilities to solve various tasks. However, we observe that the $DED^{MBe}$ method has a lower variance in transfer performance compared to $DED$. Empirical findings suggest that GAE suffers from high volatility, but overall, both methods achieve comparable performance.

### D.6 SCALABILITY OF OUR MARGINAL BENEFIT-INDUCED RL GENERATION PROCESS

We also illustrate the scalability of our marginal benefit-induced RL generation process compared with other RL-based generation approach, i.e., PAIRED and REPAIRED:

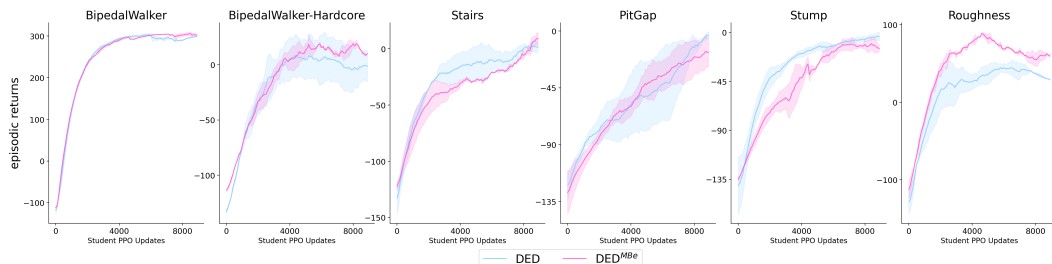

Figure 13: Performance of alternative methods for computing regret in terms of transfer performance on test environments during training (mean and standard error) over three random seeds. Here DivSP_no_SP approximates the learning potential using GAE, while DivSP_no_SP_m approximates it using the marginal benefit of level $\theta$.

[1] PAIRED often overexploits the difference in the students, leading to simple tracks that incidentally favor the antagonist (as evidenced in Jiang et al. (2021b)). Conversely, our MBeDED/MBeED approaches help mitigate this issue by recovering complexity in the generated environments, as demonstrated in Figure 11 of the Appendix, which showcases examples of car-racing tracks generated. By adopting a student-centric perspective, we focus on enhancing the student's policy within a generated environment. This approach allows us to generate environments at an appropriate pace and provides a more accurate representation of the student's actual improvement during training.

[2]In PAIRED, there are two agents: Protagonist and Antagonist. For the selected environment, the antagonist agent is trained to maximize the defined regret value, which should be the difference between the optimal antagonist's performance and the protagonist's performance, and the antagonist agent is trained to maximize the regret value (In the PAIRED experiment, for each selected level, the protagonist is updated 5 times and antagonist is updated 8 times). PAIRED further approximates the regret value by computing the difference between the maximum reward achieved by the Antagonist and the average reward attained by the Protagonist across all trajectories. While this approach is well-conceived, it can still be hindered by the inherent challenge of accurately approximating the regret value, which impacts both performance and training efficiency. Contrastingly, our marginal definition sidesteps these challenges, allowing for a more streamlined and scalable approach. By focusing on the difference in policy performance, we eliminate the need for additional training iterations and the approximation of regret values. This leads to a more efficient training process, ultimately enhancing scalability and training speed in comparison to the PAIRED approach. Our approach reduces the overall training time required for the same number of training episodes while achieving significantly better performance. When compared to PAIRED and REPAIRED (PAIRED version with a level replay buffer), MBeED emerges as the fastest option. We established PAIRED as the baseline (with a value of 1 for each environment), where a higher value indicates faster training (less time). As indicated in Table 2, MBeED outperforms both the PAIRED and REPAIRED approaches. Even with the incorporation of the diversity measurement, MBeDED maintains faster training compared to PAIRED and REPAIRED, further underscoring the scalability of our RL generation process.

Table 2: Comparisons of training speed for different approaches. A higher value indicates faster training.

| Environments Approaches | *MBeDED* | *MBeED* | *PARIED* | *REPAIRED* |
|---|---|---|---|---|
| BipedalWalker | **1.15** | **1.42** | 1 | - |
| Maze | **1.13** | **1.38** | 1 | 0.85 |
| CarRacing | **1.46** | **2.12** | 1 | 0.73 |

Table 3: The environment design space used for the BipedalWalker environment. Where values are shown as ranges, both the lower and upper bounds of the range are learned parameters. When creating a level, the size of each obstacle is sampled from the range. This is also the same as work by Parker-Holder et al. (2022)

|  | Stump Height | Stair Height | Stair Steps | Roughness | Pit Gap |
|---|---|---|---|---|---|
| Easy Init | [0,0.4] | [0,0.4] | 1 | Unif(0.6) | [0,0.8] |
| Max Value | [5,5] | [5,5] | 9 | 10 | [10,10] |

## E  EXPERIMENT DETAILS AND HYPERPARAMETERS

In this section, we provide details on the experiment setup, such as the network architecture and the devices used for training. We utilized the open codebase available at https://github.com/facebookresearch/dcd to build our implementation. All the models were trained on a single NVIDIA GeForce RTX 3090 GPU and 16 CPUs.

### E.1  ENVIRONMENT DETAILS

We follow the PAIRED approach (Dennis et al., 2020) using the same RL generator structure. For example, in the partially observable navigation task (maze environment), the teacher needs to build the environments by choosing the location of the obstacles, the goals, and the starting location of the agent. The input to the teacher (or the teacher's observation) consists of a fully observed view of the environment state, the current timestep $t$, and a random vector $z \sim \mathcal{N}(0, \boldsymbol{I}), z \in \mathbb{R}^D$ sampled for each episode. At each timestep, the teacher outputs the location where the next object will be placed; at timestep 0 it places the agent, 1 the goal, and every step afterwards an obstacle. For the Bipedalwalker environment, the input is the previous environment parameters and the random vector.

#### E.1.1  BIPEDALWALKER

We use the modified BipedalWalkerHardcore environment from OpenAI Gym for our experiments in this study. The agent is trained using a proprioceptive state consisting of 24 dimensions, which includes lidar sensors, angles, and contact information. However, the agent does not have access to its map coordinates. The action space consists of four continuous values that control the torques of its four motors.

To measure the transfer performance, we evaluate the agents on each of the individual challenges generated by the environment parameterization. In particular, we utilize the following six environments, of which five are also employed by Parker-Holder et al. (2022):

- Stairs: the stair height parameters are set to [2,2] with the number of steps set to 5.
- PitGap: the PitGap parameter is set to [5,5].
- Stump: the Stump parameter is set to [2,2].
- Roughness: the Roughness parameter is set to 5.
- BipedalWalker: the vanilla BipedalWalker without Stairs, PitGap, Sump and Roughness.
- BipedalWalker-Hardcore: the hard BipedalWalker with a mix of the above-mentioned elements.

Each of these is visualized in the main body of the paper, in Figure 3.

When generating new environments, the environment design space is shown in Table 3.

#### E.1.2  MINIGRID

The experiments in this work utilize a $15 \times 15$ discrete grid as the Partially Observable Navigation environment, where a border of walls surrounds the grid. This creates a total of $13 \times 13 = 169$ free tiles that the teacher (environment generator) can use to place obstacles, while other tiles may contain navigable space, the agent, a block, or the goal. The student agents, Alice and Bob, have

partial observability, as they are only able to perceive a $5 \times 5$ view in front of them. They are also aware of their orientation, and can only move forward or turn left/right. If the student agent encounters an obstacle, it will remain stationary. The student agent is rewarded with $1 - T/T_{max}$ upon reaching the goal, where $T$ is the episode length and $T_{max}$ is set to 250 as the maximum episode length. If the agent fails to reach the goal, it receives a reward of $0$.

The procedure for designing the environment is as follows: at each timestep, the teacher (environment generator) receives a complete map of the level as an observation and takes an action of dimensionality 169 to indicate the location of the next object to be placed. The adversary agent places objects in a sequence of actions, with the agent and goal placed on the first and second steps, respectively. The adversary then places 50 walls (obstacles) for 50 steps after the goal has been placed. If the adversary places an object on top of a previously existing object, its action does nothing, allowing it to place fewer than 50 obstacles. If it attempts to place the goal on top of the agent, the goal will be placed randomly. This procedure is similar to recent works such as those by Dennis et al. (2020); Jiang et al. (2021b). Some examples of generated maze environments are visualized in the main body of the paper, in Figure 6.

The teacher (environment generator) constructing the environment receives observations that consist of a $15 \times 15 \times 3$ image of the environment state, a current timestep integer $t$, and a random vector $z \sim N(0, I)$, where $z \in R^{50}$ to allow for the generation of random mazes. The order of actions is important, so an RNN is experimented with to parameterize the adversary, although it is not always necessary. The teacher architecture is similar to that of the student agents, with a single convolutional layer connecting to an LSTM, then to two fully connected layers connecting to policy outputs. Additional inputs like t and z connect directly to the LSTM layer. A second identical network is used to estimate the value function.

### E.1.3 CARRACING

In the CarRacing environment, we design each track as a closed loop for the student agent to drive around, with the goal of completing a full lap. To enhance the expressiveness of the original CarRacing environment, we reparametrize the tracks using Bézier curves. Specifically, each track is constructed from a Bézier curve (Mortenson, 1999) based on 12 control points randomly sampled within a fixed radius, $B/2$, of the center of the $B \times B$ playfield. The track is comprised of a sequence of $L$ polygons, with the student receiving a reward of $1000/L$ for driving over each previously unvisited polygon. In addition, the student receives a penalty of $-0.1$ at each time step. Following the methodology of (Jiang et al., 2021a), we do not penalize the agent for driving out of bounds but terminate the episode if it deviates too far off track. To reduce the complexity of the observation space, we provide the student with a $96 \times 96 \times 3$ pixel observation in RGB channels, clipped to an egocentric bird's-eye view of the vehicle centered horizontally in the top $84 \times 96$ portion of the frame. The remaining $12 \times 96$ portion of the frame displays the dashboard, which visualizes the agent's latest action and return.

The procedure for designing the environment is as follows: The teacher agent (environment generator) starts by generating a sequence of 12 control points, one for each time step, within a fixed radius of $B/2$ from the center of the playfield. The agent always starts at the polygon on the track closest to $0^o$ relative to the center and faces counterclockwise. Specifically, The teacher policy receives input at each time step consisting of the set of all previously generated control points, the current time step represented as a one-hot vector, and a 16-dimensional random noise vector. These control points are spatially represented in a $10 \times 10$ grid called the sketch, which serves as a downsampled and discretized version of the playfield boundaries where the track is generated. Each control point is selected by choosing a cell within the grid, ensuring that no two control points are placed too close together, which would result in excessive track overlapping. Finally, the chosen control point's cell coordinates are upscaled to match the original playfield scale. For more details, please refer to Jiang et al. (2021a)

### E.2 HYPERPARAMETERS

We made use of the majority of hyperparameters from previous works like (Dennis et al., 2020; Jiang et al., 2021b;a), with minor modifications. For BipedalWalker, we employed the continuous control policy from the open-source implementation of PPO presented in Kostrikov (2018), along

with most of the recommended hyperparameters for MuJoCo. The policy uses a simple feedforward neural network with two hidden layers of size 64 and Tanh activations.

Regarding Minigraid, we followed the setup described in Dennis et al. (2020), which utilizes a teacher agent with 128 convolution filters, an entropy regularization coefficient of 0.0, and a student episode length of 250. All agents employed a convolution kernel size of 3, an LSTM size of 256, two fully connected layers of size 32 each, and a fully connected layer of size 10, which inputs the timestep and connects to the LSTM.

For CarRacing, in order to encode the sketch mentioned above, two $2 \times 2$ convolutions with a stride length of 1 are used, with 8 and 16 channels respectively, each followed by a ReLU layer. The flattened outputs from these convolutions are concatenated with an 8-dimensional embedding of the time step and a 16-dimensional random noise vector. This combined embedding is then fed through two fully connected layers, with a hidden size of 256 for each, where the first layer is followed by a ReLU activation, resulting in policy logits over the 100 possible control point choices. It should be noted that any cells in the sketch that have already been selected are masked out to prevent duplicate selection of the same control point.

The student agent is composed of an image embedding module that uses a stack of 2D convolutions with square kernels of sizes $2, 2, 2, 2, 3, 3$, channel outputs of $8, 16, 32, 64, 128, 256$, and stride lengths of $2, 2, 2, 2, 1, 1$ respectively. The resulting image embedding is 256-dimensional. This embedding is then processed by a fully connected layer with a hidden size of 100, followed by a ReLU layer. The output of the ReLU layer is then passed through two separate fully-connected layers, each with a hidden size of 100 and output dimension equal to the action dimension. Softplus activations are applied to the output of each fully connected layer, and we add 1 to each component of the resulting two output vectors. These vectors serve as the $\alpha$ and $\beta$ parameters respectively for the Beta distributions used to sample each action dimension. During training, rewards are normalized by dividing rewards by the running standard deviation of the returns encountered so far.

For a list of hyperparameters for each experiment please see Table 4. Additionally, some of the hyperparameters for the baselines can be found in references such as (Dennis et al., 2020; Jiang et al., 2021b;a; Parker-Holder et al., 2022).

Table 4: Hyperparameters used for training each method in the BipedalWalker, Minigrid, and Car-Racing environments

| Parameter | BipedalWalker | MiniGrid | CarRacing |
|---|---|---|---|
| PPO | | | |
| $\gamma$ | 0.99 | 0.995 | 0.99 |
| $\mathbf{\Lambda}_{gae}$ | 0.9 | 0.95 | 0.9 |
| PPO rollout length | 2048 | 256 | 125 |
| PPO epochs | 5 | 5 | 8 |
| PPO mini-batches per epoch | 32 | 1 | 4 |
| PPO clip range | 0.2 | 0.2 | 0.2 |
| PPO number of works | 16 | 32 | 16 |
| PPO update | 10000 | 20000 | 1800 |
| Adam learning rate | 3e-4 | 1e-4 | 3e-4 |
| Adam $\epsilon$ | 1e-5 | 1e-5 | 1e-5 |
| PPO max gradient norm | 0.5 | 0.5 | 0.5 |
| PPO value clipping | no | yes | no |
| return normalization | yes | yes | yes |
| value loss coefficient | 0.5 | 0.5 | 0.5 |
| student entropy coefficient | 1e-3 | 0.0 | 0.0 |
| generator entropy coefficient | 1e-2 | 0.0 | 0.0 |
| scoring function | positive value loss | positive value loss | positive value loss |
| replay probability | 0.5 | 0.5 | 0.5 |
| buffer size, $K$ | 128 | 256 | 256 |
| level replay score transform | rank | rank | power |
| Temperature, $\beta$ | 0.1 | 0.3 | 1 |
| diversity coefficient | 0.5 | 0.3 | 0.5 |
| diversity score transform | power | power | power |

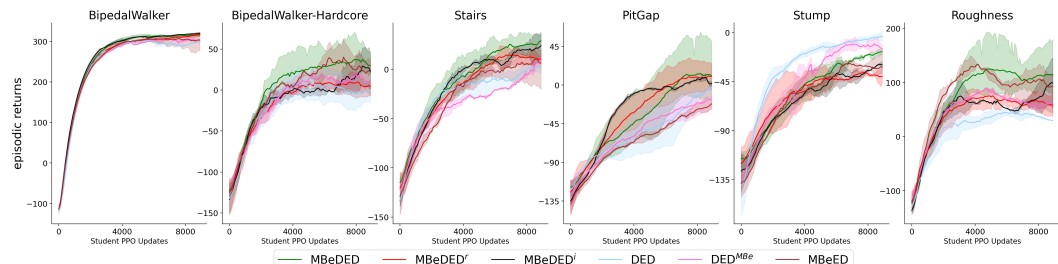

Figure 14: Performance on test environments during training In the BipedalWalker environment. Here *MBeDED*$^r$ considers only representative states when selecting $S_\theta$ for level $\theta$, while *MBe-DED*$^i$ considers only the important observed states. DED and DED$^{MBe}$ use a randomly generated environment teacher, and DED approximates the learning potential using GAE, DED$^{MBe}$ approximates it using the marginal benefit of level $\theta$. *MBeED* is a version only considers marginal benefit.

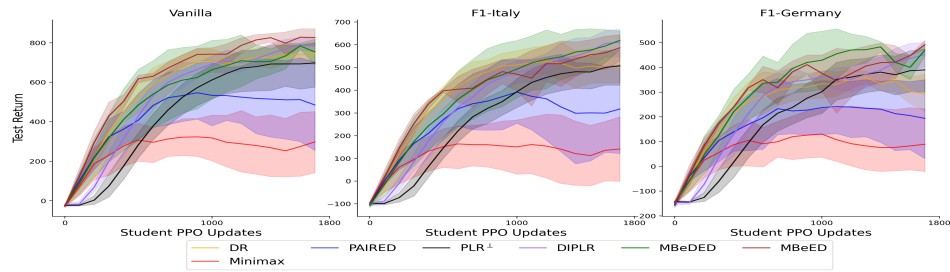

Figure 15: Zero-shot transfer performance on the OOD F1 tracks: Vanilla, Italy and Germany. We show that MBeED demonstrates strong performance when compared to the baseline methods.

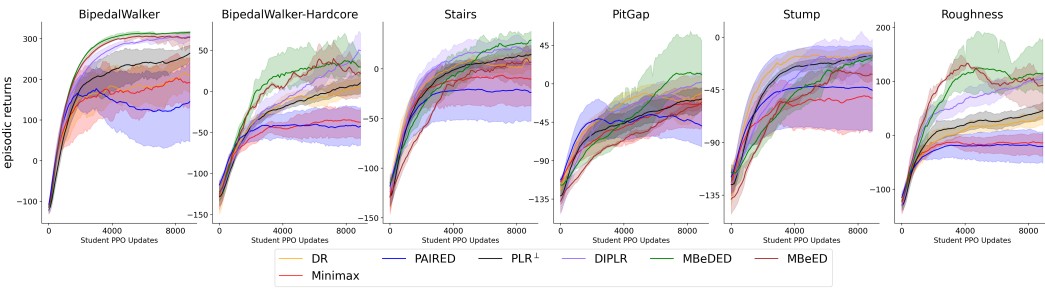

Figure 16: Performance on test environments during training. *MBeDED* against other baselines.

