# OpenReview forum: "Marginal Benefit Induced Unsupervised Environment Design"
_ICLR.cc/2024/Conference — ICLR 2024 Conference Withdrawn Submission_

### Official Review · Reviewer_32af · 2023-10-26

**Soundness:** 3 good
**Presentation:** 1 poor
**Contribution:** 2 fair
**Rating:** 3
**Confidence:** 4

**Summary:**

The paper introduces MBeDED, an unsupervised environment design (UED) method, that uses "marginal benefit" as a proxy of how much an RL agent (student) can improve on the generated environment. In addition to that, MBeDED uses 'representative score' when adding environments to the replay buffer to increase the novelty of the environments the agent is trained on.

**Strengths:**

* I really like the proposed way of estimating the 'marginal benefit', and the simplicity of it.
* The method significantly outperforms the existing approaches on some of the environments.

**Weaknesses:**

* The paper is very hard to read/understand. It took me several passes to grasp what's going on.
	* Already in the introduction, the authors use the terms that were not yet introduced, e.g. regret, GAE etc.
	* The notation is often confusing. Some sets are defined using \mathbb, some of them are capital bold letters. The 'representative observed state vector' is, in reality, not a vector, but a set.
	* Adding the parameterization to the models used in the paper (policies, value functions etc) would also help readability/understanding. Knowing what input / reward goes to which agent and what models each agent uses would greatly help the reader.
* The paper focuses on the results more than on understanding.
	* In my opinion, the paper would be much more interesting and useful to the community if it goes deeper trying to understand the effect of the algorithmic decisions taken by the authors. For example, the paper poses an interesting hypothesis: "While regret represents the learning potential of an environment, it fails to indicate whether all that potential can be achieved by a student and if so in how many steps." Validating this hypothesis empirically (and comparing regret vs marginal benefit) would be extremely interesting. This will also be practically useful as we want to know which design decisions to make and what limitations to take into account outside of the world of the BipedalWalker and Minigrid.

**Questions:**

* As I mentioned in the Weaknesses section above, I highly recommend the authors to define all the concepts before they use them in the paper.
* I would appreciate the authors addressing the notation issues I mentioned in the 'Weaknesses' section.
* In your opinion, what are the implicit assumptions you made when designing MBeDED?
* You treat each task as a POMDP, however, in Section 3.2, the value function takes states as inputs, and the diversity metrics are calculated on the states, not on the observations. Do you assume access to the states here? What are the states? Or do you calculate  those on observations?
* I think the amount of mathematical notation can be significantly reduced in Section 3.2 as it makes it harder to read, and I don't think some of it is needed. The notation of the F_{div} is a bit inconsistent. Sometimes it takes two sets as inputs (e.g. RHS of equation 8), but sometimes it takes a vector as a first argument and a set as a second.
* Can you think of any experiments to identify quantitative/qualitative difference between using regret vs marginal benefit? Could you think of any experiments to understand better why in some environments MBeED is not significantly different from MBeDED, but in some it is (e.g. in PitGap). What's so different between BipedalWalker and MiniGrid that makes DR~MBeDED on MiniGrid, but DR < MBeDED on some environments in Figure 4?
* Some of the methods in Figure 4 does not seem to converge yet. Do you think asymptotically MBeDED and PLR will reach the same point? What happens if we train PLR longer for BipedalWalker-Hardcore?
* Personally, I would get rid of Alice/Bob in the algorithm explanation and just use a time subscript to denote the policies in question. In my opinion, introducing more entities complicates the narrative.

---

### Official Review · Reviewer_vj8C · 2023-10-30

**Soundness:** 2 fair
**Presentation:** 2 fair
**Contribution:** 2 fair
**Rating:** 5
**Confidence:** 4

**Summary:**

The paper proposes MBeDED, a PAIRED-style algorithm for unsupervised environment design. A measure of "marginal benefit" is used to prioritize environments, which computes how much a policy is improved on one environment by training on another environment. A heuristic measure of diversity is proposed to increase the diversity of environments in the replay buffer. Experiments on simple tasks show improvement over prior work.

**Strengths:**

- The paper addresses an interesting and relevant problem of unsupervised environment design, which is very challenging for current algorithms
- The proposed method is intuitive and intriguing
- The experiments suggest the method improves over prior work

**Weaknesses:**

- The proposed diversity metric is entirely heuristic and relies on kernel distance between states, which will be challenging to scale to more complex environments.
- The method lacks solid theoretical foundation.
- The experiments are only performed in very simple environments
- The proposed diversity objective is not ablated. It is unclear how important it is for performance

**Questions:**

1. I would like to see the proposed diversity objective ablated. Even better would be to implement baselines using the diversity objective.
2. The method is only evaluated on simple environments. It would be good to add more complex continuous control environments, e.g. continuous control maze or antmaze.

---

### Official Review · Reviewer_QN2d · 2023-10-31

**Soundness:** 3 good
**Presentation:** 2 fair
**Contribution:** 2 fair
**Rating:** 3
**Confidence:** 3

**Summary:**

This paper proposes a UED method that combines existing generative and curation methods, with a novel marginal benefit objective for training the generator and diversity metric for buffer curation. The performance of the method is evaluated on BipedalWalker, MiniGrid, and Car Racing environments against a range of prior approaches.

I am recommending rejection for this paper, primarily due to the evaluation, but also due to practical concerns with the method. The presentation of the results makes it impossible to determine a significant improvement from the method, and the experiments are underpowered (5 seeds) given the variance in the results.

**Strengths:**

1. The motivation and description of the proposed method is detailed.
2. The proposed method is evaluated against many prior UED methods and covers a wide range of environments.
3. The ablation experiment in Figure 5 is an effective approach to demonstrating the contribution of the components, however the plots are very hard to parse and appear to show an insignificant difference from all ablations.

**Weaknesses:**

1. The proposed environment generation objective, marginal benefit, seems to be practically flawed. It appears the agent from one update ago is used as the baseline, which is unlikely to be significantly different from the updated agent given the total number of updates (100k). Due to this, marginal benefit likely just approximates the variance in the cumulative return of the agent, returning high values when the updated agent happens to randomly sample a better trajectory. I would like to understand why this is not the case if the previous agent is used as the baseline, or how many updates are used if I am mistaken.
2. As the state-of-the-art method on these UED domains, this method should definitely be evaluated against ACCEL. This is particularly pertinent as ACCEL is the most similar to this method, combining a generator strategy with curation. The reason for not evaluating ACCEL is missing, presumably due to a typo (bottom of page 7).
3. Table 1 in the appendix is clearly inspired by Table 3 in the reference ACCEL publication (Parker-Holder et al., 2020), but is uncredited and omits ACCEL. This is borderline plagiarism, making it even more unclear why the method does not compare to ACCEL.
4. It is difficult to draw any conclusions from the evaluation due to the majority of results being presented as training curves with no significance testing. The majority of these curves appear to show insignificant differences due to very high variance in performance, whilst on the plot presenting final performance median and IQR (Figure 7), the proposed method is insignificantly different to DR on every level (although this is not possible to establish exactly) and appears insignificantly different to PLR on the majority of levels. It would be very beneficial to see more seeds and significance testing based on final performance.
5. There is no discussion or analysis of the Car Racing experiments, presumably due to a lack of space.
6. The quality of the figures is lacking. Figure 1 is difficult to parse, the plot format in Figure 2 is inconsistent with the later plots, and Figures 3 and 6 add little information to the paper, so should be moved to the appendix to make room for e.g. the Car Racing evaluation.

**Questions:**

1. Is marginal benefit computed against the agent before the most recent update, or are multiple updates used?

---

### Official Review · Reviewer_gp3R · 2023-11-01

**Soundness:** 2 fair
**Presentation:** 3 good
**Contribution:** 3 good
**Rating:** 5
**Confidence:** 5

**Summary:**

The paper presents a novel approach, MBeDED, that utilizes marginal benefit and diversity to enhance the UED process. This method involves two student agents and a teacher, where the teacher generates levels that are challenging yet feasible, and also aims to curate more diverse levels in comparison to those curated by existing approaches.

**Strengths:**

- The paper introduces a novel approach, incorporating two new methods to address UED.
- The algorithm section is comprehensive, detailing all significant aspects of the method.

**Weaknesses:**

1. The paper advocates the use of marginal benefit over regret, an already established and theoretically validated metric, without providing a substantial discussion, either theoretically or empirically, supporting why marginal benefit is better than regret.

2. Given that the test results are similar to prior works in Minigrid and CarRacing, a discussion comparing this method with PAIRED, and identifying any shortcomings of PAIRED that MBeDED addresses, would be beneficial. Prior research, such as [1] provides a thorough analysis in this area.

[1]: Mediratta, I., Jiang, M., Parker-Holder, J., Dennis, M., Vinitsky, E., & Rocktäschel, T. (2023). Stabilizing Unsupervised Environment Design with a Learned Adversary. arXiv preprint arXiv:2308.10797.

3. The results in CarRacing are only presented for three test tracks, in contrast to the twenty tracks used in PLR. A broader evaluation, including all test tracks in PLR, would better validate the robustness of the method. Similar is the case for Minigrid as well.

4. As MBeDED's performance improves significantly when the diversity metric is applied, it would be advisable to move the DIPLR results from the appendix to the main section, and provide a detailed analysis on this aspect.

In its current state, the paper requires further analysis to convincingly argue why MBeDED should be preferred over existing methods like PAIRED or PLR. Additionally, the results presented appear selective/cherry-picked, necessitating a more comprehensive analysis to establish the method's robustness, to make this submission suitable for acceptance at this conference.

**Questions:**

On page 7 (second last line), there seems to be an incomplete sentence that is crucial for understanding the lack of comparison against ACCEL.

> "… as it re We show the average and variance of the performance for our method, baselines with five random seeds."

---

### Official Review · Reviewer_oFUv · 2023-11-03

**Soundness:** 3 good
**Presentation:** 2 fair
**Contribution:** 2 fair
**Rating:** 3
**Confidence:** 4

**Summary:**

This paper discusses the challenges of training generally capable Reinforcement (RL) agents in complex environments, and critiques current Unsupervised Environment Design (UED) approaches that use regret-based objectives for adaptive curriculum learning. The paper introduces an alternative objective based on marginal benefit, which reflects the real improvement an agent experiences from an environment. They also propose a new scalable RL-based method for controlled environment generation that leverages this marginal benefit objective along with a novel concept of environment diversity. This method is posited to avoid issues like catastrophic forgetting and enable faster convergence of the agent's learning process. The paper's key contributions include defining marginal benefit for environment generation, introducing a scalable RL-based teacher for environment generation, and proposing a new mechanism for training on diverse environments.

**Strengths:**

- Shortcomings of existing UED methods are well described and explained, and possible solutions in addressing them are well motivated in the paper.
- The idea of using marginal benefits as the objective for teacher in UED is intuitive and novel.
- The idea behind the representative states is also interesting and novel AFAIC, specifically in applications in curriculum learning.

**Weaknesses:**

### Experimental Results

My primary criticism regarding the work lies in the provided results, which do not align with those presented in prior papers. For instance, DR performs better than Robust PLR in 5 out of 6 MiniGrid environments. This directly contradicts the results in the PLR paper [1], presented at NeurIPS 2021 and made open-source since July 2022, casting doubts on the accuracy of all experiments presented in this work.
- One notable difference is the number of steps for which the authors ran the experiments. For example, the MiniGrid results in the PLR are for 250 million steps, while those presented here are for only 100 million steps. This could potentially affect Robust PLR more, given it's conducting half the number of PPO updates.
- Another discrepancy is in the number of seeds used. In the original paper, the authors use 10 seeds, whereas here it is reduced to 5, introducing more noise into the results.
- Selective presentation of environments raises questions as well. It is unclear to me why the authors present results on Maze but omit results on Maze 2. Similarly, why only present results on 3 Formula 1 tracks rather than the entire set?

Note: I completely understand that running these experiments for longer and with a higher number of seeds is computationally expensive. However, I believe it is important to adhere to fixed evaluation protocols for related works in the same field to allow for apples-to-apples comparisons where possible.

### Other issues

- There is prior work on utilizing diversity within the context of UED. Specifically, [2] adaptively identifies diverse environments using the Wasserstein distance measure, which is relevant to environment design. This work is not mentioned or compared against.
- Why isn't the method compared to ACCEL? The authors provide only half a sentence on this topic, but the second half of the sentence is missing. I can surmise their reason might be that ACCEL uses editing; however, one could argue that the newly proposed framework could also incorporate editing. If that's the case, there would be an expectation to explore this possibility and compare the results.
- What exactly is the MBeED approach? This term is first introduced in Section 3.2 as if it has already been explained, yet there is no previous mention of it. If it shares common procedures in the pseudocode with MBeDED, then this should be clearly stated, and potentially highlighted with color-coding in Algorithm 1.
- The lines of pseudocode in Section 3.1 are incorrect. For instance, Line 10 does not correspond to the creation of new environments. It seems the authors wrote this section and then updated the pseudocode. Regardless, this needs to be fixed as it creates confusion."

Given the major issues in this work, particularly regarding experimental results, I recommend that the paper is rejected in its current form. That said, I am looking forward to discussing this work with the authors during the rebuttal.

[1] Jiang el al, ****Replay-Guided Adversarial Environment Design****, https://arxiv.org/abs/2110.02439

[2] Li et al, **Generalization through Diversity: Improving Unsupervised Environment Design**, https://arxiv.org/abs/2301.08025

**Questions:**

Please take a look at the weaknesses above and address them.